# An Analysis of Vertebral Body Growth after Proton Beam Therapy for Pediatric Cancer

**DOI:** 10.3390/cancers13020349

**Published:** 2021-01-19

**Authors:** Keiichiro Baba, Masashi Mizumoto, Yoshiko Oshiro, Shosei Shimizu, Masatoshi Nakamura, Yuichi Hiroshima, Takashi Iizumi, Takashi Saito, Haruko Numajiri, Kei Nakai, Hitoshi Ishikawa, Toshiyuki Okumura, Kazushi Maruo, Hideyuki Sakurai

**Affiliations:** 1Proton Medical Research Center, Department of Radiation Oncology, University of Tsukuba Hospital, Tsukuba, Ibaraki 305-8576, Japan; baba@pmrc.tsukuba.ac.jp (K.B.); ooyoshiko@pmrc.tsukuba.ac.jp (Y.O.); shimizu@pmrc.tsukuba.ac.jp (S.S.); nakamura@pmrc.tsukuba.ac.jp (M.N.); hiroshima@pmrc.tsukuba.ac.jp (Y.H.); iizumi@pmrc.tsukuba.ac.jp (T.I.); saitoh@pmrc.tsukuba.ac.jp (T.S.); haruko@pmrc.tsukuba.ac.jp (H.N.); knakai@pmrc.tsukuba.ac.jp (K.N.); ishikawa.hitoshi@qst.go.jp (H.I.); okumura@pmrc.tsukuba.ac.jp (T.O.); hsakurai@pmrc.tsukuba.ac.jp (H.S.); 2Department of Radiation Oncology, Tsukuba Medical Center Hospital, Tsukuba, Ibaraki 305-8558, Japan; 3National Institutes for Quantum and Radiological Science and Technology, QST Hospital, Chiba 263-8555, Japan; 4Department of Clinical Trial and Clinical Epidemiology, Faculty of Medicine, University of Tsukuba, Tsukuba, Ibaraki 305-8575, Japan; maruo@md.tsukuba.ac.jp

**Keywords:** proton beam therapy, pediatric, child, bone, vertebral body

## Abstract

**Simple Summary:**

Radiotherapy has a key role in treatment of pediatric cancer and has greatly improved survival in recent years. However, vertebrae are often included in the irradiated area, and this may affect growth after treatment. In this study, we examined the relationship of the dose of proton beam therapy with subsequent growth of 353 vertebral bodies in 23 children (10 boys, 13 girls) with a median age at treatment of 4 years old and a median observation period of 13.9 months. Most importantly, we found that the growth rate of vertebral bodies decreased even at a low proton beam therapy dose, which indicates the need for careful planning of the irradiation area in this patient population. Growth inhibition was clearly dose-dependent, and proton beam therapy had the same growth inhibitory effect as photon radiotherapy, at least within the irradiated field.

**Abstract:**

Impairment of bone growth after radiotherapy for pediatric bone cancer is a well-known adverse event. However, there is limited understanding of the relationship between bone growth and irradiation dose. In this study, we retrospectively analyzed bone growth impairment after proton beam therapy for pediatric cancer. A total of 353 vertebral bodies in 23 patients under 12 years old who received proton beam therapy were evaluated. Compared to the non-irradiated vertebral body growth rate, the irradiated vertebral body rate (%/year) was significantly lower: 77.2%, 57.6%, 40.8%, 26.4%, and 14.1% at 10, 20, 30, 40, and 50 Gy (RBE) irradiation, respectively. In multivariate analysis, radiation dose was the only factor correlated with vertebral body growth. Age, gender, and vertebral body site were not significant factors. These results suggest that the growth rate of the vertebral body is dose-dependent and decreases even at a low irradiated dose. This is the first report to show that proton beam therapy has the same growth inhibitory effect as photon radiotherapy within the irradiated field.

## 1. Introduction

Radiotherapy plays an important role in multidisciplinary treatment for various childhood tumors. Recent progress in treatment methods has increased the overall 5-year survival rate of childhood tumors to over 80% [1]. Thus, the long-term effects of radiotherapy, such as secondary cancer and other late adverse events, have become important concerns for long-term survivors. Mortality due to the original malignancy begins to plateau beyond 20 years, while death from other causes increases [2]: from 15 to 30 years after diagnosis of a pediatric cancer, the cumulative mortality attributable to the primary disease only increases from 6.3% to 7.8%, while that due to non-recurrence causes increases from 2.0% to 7.0%. Vertebrae are often included in the irradiation field in radiotherapy for a pediatric tumor, especially for tumors near the spine, such as neuroblastoma or nephroblastoma, and patients who are at risk of meningeal dissemination such as medulloblastoma or ependymoma may receive craniospinal irradiation (CSI) [3]. Many studies have analyzed bone growth after radiotherapy for pediatric patients and found that short stature is common after CSI [4]. Hartley et al. found impaired vertebral growth among survivors of medulloblastoma and supratentorial primitive neuroectodermal tumors (follow-up; 44 months) who received 36–39.6 Gy CSI compared to those who received 23.4 Gy. Probert et al. found that sensitivity to irradiation is higher in patients under 6 years of age and at puberty [5]. They evaluated twenty-two patients under the age of 15 years, who received approximately 40 Gy of complete spinal irradiation with a linear accelerator for treatment of lymphoma, medulloblastoma, and acute lymphoblastic leukemia. In addition, 16 of 22 children showed retardation of growth of the spine. However, studies on bone growth after radiotherapy have not used a clearly defined threshold for short stature [6]. Based on limited available information, they advised that homogeneous vertebral radiotherapy doses should be delivered to children who have not yet finished the pubertal growth spurt. It is recommended that vertebral delineation should include all primary ossification centers and growth plates. For partial spinal irradiation, they also recommended that the number of irradiated vertebrae should be restricted as much as possible, particularly at the thoracic level in young children (<6 years old). Proton beam therapy is likely to reduce the risks of growth and development disorders, endocrine dysfunction, reduced fertility, and secondary cancer in children. This is because of its characteristic distribution of a decreased dose and volume in normal organs, and maintenance of efficacy against the tumor [7,8,9]. Several reports have shown the merit of proton beam therapy for cancers in children, focusing on secondary cancer [10,11,12,13,14]. Sethi et al. evaluated 86 pediatric patients with retinoblastoma who were treated with proton beam therapy (*n* = 55) or photon radiotherapy (*n* = 31). They found that the 10-year cumulative incidence of radiotherapy-induced or in-field secondary malignancies was significantly different between radiation modalities (proton vs. photon: 0% vs. 14%; *p* = 0.015). The 10-year cumulative incidence of all secondary malignancies was also different, although there was no significance (5% vs. 14%; *p* = 0.120). However, there was no information about bone growth after proton beam therapy in children. In this retrospective study, we examined quantitative relationships between dose and vertebral growth changes after proton beam therapy.

## 2. Results

Of the 353 vertebral bodies examined in 23 patients, 61, 211, and 81 were cervical, thoracic, and lumbar vertebrae, respectively. At the start of proton beam therapy, the median lengths of the cervical, thoracic, and lumbar spine were 6.7 (3.6–9.2), 10.9 (6.2–17.9), and 14.2 (10.4–18.6) mm, respectively. The median observation period was 13.9 months (9.4–19.1 months). The respective median growth rates after proton beam therapy at 0 (no irradiation of the vertebral body), 20, and 30 GyE were 6.47 (0–40.4%), 5.80 (1.9–19.4%), and 2.40 (1.3–6.8%) for cervical vertebral bodies; 7.23 (0–30.0%), 3.61 (0–5.7%), and 2.10 (0–11.6%) for thoracic vertebral bodies; and 7.12 (0–30.1%), 4.47 (0–14.0%) and 2.70 (0–8.8%) for lumbar vertebral bodies. At all three sites, the growth rate became lower as the irradiation dose increased (Figure 1, Figure 2 and Figure 3).

The growth rate for each patient is shown in Figure 4. In all patients, the growth rate tended to decrease linearly with the irradiation dose. In statistical analysis using the linear mixed model, age and gender were not selected as fixed effects in predicting the effects of dose. The decrease in growth rate due to the increase in dose was highly significant (*p* < 0.0001). The median growth rate of the cervical spine was slightly higher than that of other sites (*p* = 0.0071), but the interaction between the site and dose was not selected as a fixation effect. It was not suggested that the slope of the growth rate curve would change depending on the site. The result of the fixed effects estimation with a best-subset selection is shown in Table 1.

There was no specific dose threshold at which the growth rate changed rapidly. Therefore, a simple model using the proton dose only was selected for predicting vertebral body growth deficit. The median curve and 2.5 and 97.5 percentile prediction curves for the growth rate and dose are shown in Figure 5. The equation for the median curve of the growth rate in Figure 5 is “One-year growth rate = exp (2.59 − 0.0154 × *dose*) − 5 (%/year)”. Compared with a non-irradiated vertebral body, the growth and growth rate (%/year) were 7.34 (88.1% compared with the non-irradiated vertebral body), 6.43 (77.2%), 5.58 (67.0%), 4.80 (57.6%), 4.07 (48.9%), 3.40 (40.8%), 2.78 (33.3%), 2.20 (26.4%), 1.67 (20.0%), 1.17 (14.1%) at 5, 10, 15, 20, 25, 30, 35, 40, 45, and 50 GyE irradiation, respectively.

## 3. Discussion

In Japan, proton beam therapy is mainly used for liver cancer, lung cancer, esophageal cancer, and prostate cancer in adults [15,16,17,18,19]. Radiotherapy for hepatocellular carcinoma was previously considered only for palliative treatment. However, proton beam therapy could achieve high local control without severe toxicity, 5-year local control was about 80–90%, and proton beam therapy is being considered as curative treatment. In many cases, dose escalation has been tried to achieve better local control compared to photon radiotherapy [20,21,22]. In contrast, proton beam therapy for children was mainly used to reduce adverse events. Several reports have indicated that proton beam therapy has the potential to reduce late toxicity while maintaining the same treatment effect as photon radiotherapy [23,24,25,26]. Hirano et al. evaluated the quality-adjusted life years (QALYs) of a model patient who received CSI at 6 years of age. The authors concluded that proton beam therapy with cochlear dose reduction improves health outcomes at a cost that is within the acceptable cost-effectiveness range from the payer’s standpoint [23]. Vega et al. performed a cost-effective analysis from a societal perspective using a Monte Carlo simulation model [24]. They came to a similar conclusion that proton therapy is a cost-effective strategy for the management of pediatric patients with medulloblastoma compared with standard of care photon therapy.

Recently, the American Society for Radiation Oncology model policy classified proton beam therapy for pediatric malignancy as Medically Necessary. However, proton beam therapy is still relatively new, so information about late toxicity after proton beam therapy is expected based on the experience of photon radiotherapy. Facial bone irradiation-induced facial deformation [27] and whole spine irradiation-induced short stature are changes that have a major effect on future life. Growth retardation after CSI was first suggested in 1969 by Bloom et al. [28]. Growth retardation is significantly associated with irradiation dose and age at CSI [29,30,31]. Several studies have analyzed bone growth after radiotherapy in pediatric patients [32,33,34,35], mostly in patients who received CSI. In 1975, Probert et al. found sitting height impairment in children who received whole spinal irradiation at a dose of 25 Gy [5]. Hartley et al. also reported that a high dose (36–39.6 Gy > 23.4 Gy) was a significant risk factor for growth retardation [4]. The dose for CSI usually ranges from 20 to 36 Gy, and Oshiro et al. evaluated 892 vertebral bodies in 220 pediatric patients treated with CSI. They found that vertebral bone growth was significantly worse after a dose of 39 Gy [36]. The bone growth rates were 4.0%, 3.0%, 2.5%, 2.3%, 1.2%, and 0.5% for bone receiving <20, 23.4, 36.0, 39.6, 40–50, and >50 Gy, respectively. Only a few bones received <20 or >40 Gy, so the impact of lower and higher doses was unclear. However, their results also indicated that the growth rate of the vertebral body was dose-dependent. The effects of lower doses are unclear, although a few reports suggest growth retardation at low doses. Thus, Hogeboom et al. found that 10 Gy abdominal irradiation for nephroblastoma caused growth disturbance, suggesting that bone growth is suppressed at <20 Gy [37]. In an evaluation of the results of vertebral sparing using intensity-modulated radiation therapy (IMRT) for neuroblastoma, Ng et al. reported that vertebral bodies irradiated at 12.9 Gy grew significantly slower than out-of-field controls [38].

In this study, several vertebral bodies that received different doses in one patient were evaluated. As the height of the vertebral body before proton therapy differs depending on the site, we calculated the growth rate (%/year) instead of the absolute value of the vertebral body height to evaluate the growth of the vertebral body. The results suggested that bone growth retardation occurred even at a low dose of 10 Gy and that the growth rate linearly decreased as the dose increased without reaching a threshold value. In addition, no other factors had a significant impact on bone growth, including age and sex. In contrast, Hartley et al. identified female sex and younger age as significant risk factors for bone growth retardation after CSI; Hogeboom et al. found a more severe effect on height at a lower age at the time of radiotherapy [4,37]; Mizumoto et al. suggested that age at irradiation and sex were significant factors for future stature after CSI, in addition to radiation dose [39]. Height impairment for children who received CSI younger than 10 years of age was 8.1% for males and 4.4% for females at 20 years of age. To summarize what is known at this time, spinal growth retardation is significantly worse in those who received higher dose CSI or received CSI at a younger age. Skeletal hypoplasia, craniofacial deformities, and facial bone deformities after radiotherapy are also considered dose-dependent. Many reports used a cut-off line of about 20–30 Gy, because standard CSI doses for neuroblastoma in cases of medulloblastoma and low-risk rhabdomyosarcoma are around 20–40 Gy. Therefore, the cut-off point is unclear if pediatric patients before puberty receive 20 Gy or more during radiotherapy for bone or soft tissue, dose-dependent late effects were inevitable. Therefore, we minimized the irradiation volume and performed homogeneous vertebral body radiotherapy for children who had not yet finished the pubertal growth spurt. Oshiro et al. also measured vertebral height after CSI, and suggested that radiation dose and hormone replacement therapy were the only significant factors for vertebral height growth [36], which is more consistent with our study. Assuming that bone growth continues to slow down after irradiation, our findings are also consistent with irradiation at a younger age, leading to a lower final height.

One of the limitations of this study is the short observation period, which was a maximum of 19.1 months. However, growth disturbances of the vertebral body appeared immediately after irradiation. The number of patients was small and there may also have been measurement errors. As far as we are aware, this is the first study to evaluate several vertebral bodies that received different doses in a single patient, and we found that growth retardation can occur even at low doses of 10 Gy(RBE). We consider that this information is very important. This result can be attributed to the fact that asymmetric irradiation of the vertebral body in young patients should be avoided as much as possible even at lower irradiation doses. In particular, in new treatments such as proton beam therapy and IMRT, we can freely adjust the irradiation area according to the shape of the tumor, and thus the dose distribution becomes more complicated. Therefore, we had to pay attention not only to cover the tumor enough, but also to irradiate the area that will grow in the future.

## 4. Patients and Methods

Of pediatric patients who received proton beam therapy from 2009 to 2017, we selected those who met the following criteria to assess vertebral growth: age at the start of proton beam therapy <12 years old, treatment field including a vertebral body, images of vertebrae on CT or MRI available just before proton beam therapy and about one year after the end of proton beam therapy, and comparable vertebral bodies with different irradiation doses in the imaging range. Patients who received CSI were excluded. The characteristics of the 23 eligible patients included in the study are shown in Table 2. There were 10 males and 13 females, and the median age at treatment was 4 years old (range 2–10 years). The irradiation dose ranged from 10.8 to 56.8 Gy(RBE) (median: 30.6 Gy(RBE)). The tumor types were neuroblastoma, Wilms’ tumor, Ewing sarcoma, ependymoma, and others in 13, 3, 2, 2, and 3 patients, respectively. Two patients had a head, including a pituitary gland in the field of proton beam therapy, but none required growth hormone treatment. In addition, two Wilms’ tumor patients underwent proton beam therapy after surgery. The study was approved by the Institutional Review Board (Tsukuba Clinical Research & Development Organization, H30-099).

Heights of all vertebral bodies below C3 that were fully included in the treatment field were measured using sagittal images taken pre-and post-proton beam therapy. Vertebral bodies below C3 outside the treatment field were also measured to evaluate growth of non-irradiated vertebral bodies. The height of the posterior quarter of the vertebral body was measured to avoid error from a concave vertebral body. The proton beam therapy dose to each vertebral body was also evaluated. For each vertebra, the growth rate (%/year) was calculated from the height before to that after treatment and the interval between examinations.

Proton bean therapy was basically performed 5 days per week. Treatment planning computed tomography (CT) images of the treatment site were obtained at least 3 mm intervals. Proton beam energies of 250 MeV were generated using a booster synchrotron at Proton Medical Research Center (PMRC). The treatment planning system determined the dose distributions and collimator configuration, bolus, and range-shifter thickness settings. A relative biological effectiveness (RBE) of 1.1 was assumed.

Analysis of the raw data revealed that the variance of the growth rate differed depending on the dose, and the upward variation was particularly large at 0 GyE. Therefore, we applied a linear mixed-effects model after logarithmic transformation. The linear mixed model was applied to evaluate factors that may affect vertebral growth, including log (growth rate+5) as the outcome variable; dose, square of dose, site, age, and sex as the fixed effects; and subject as the random effect. Variable selection for the fixed effects was performed using a best-subset selection procedure with AICc. Using the selected model, the mean and 2.5 and 97.5 percentile lines were estimated for the selected models and inverse transformed to the original growth rate scale (i.e., exp(x)−5). The inverse-transformed curves were regarded as the model median and 2.5 and 97.5 percentile curves, under the assumption that the residuals of the model on the transformed scale followed a normal distribution. All statistical analyses were conducted with SAS ver. 9.4 (SAS Institute Inc., Cary, NC, USA).

## 5. Conclusions

In use of proton beams, there may be a dose penumbra in one vertebral bone due to the Bragg peak. Our results show that even a low proton beam therapy dose can lead to retardation of bone growth, which indicates that careful attention to irradiation coverage considering kyphosis or scoliosis is required, even at a low dose. A further study is needed to evaluate the effect of the dose penumbra of proton beams in pediatric patients. Three important results were obtained in this study: the growth rate of vertebral bodies decreased, even at a low proton beam therapy dose; the growth inhibition rate was clearly dose-dependent; and proton beam therapy had the same growth inhibitory effect as photon radiotherapy, at least within the irradiated field.

## Figures and Tables

**Figure 1 cancers-13-00349-f001:**
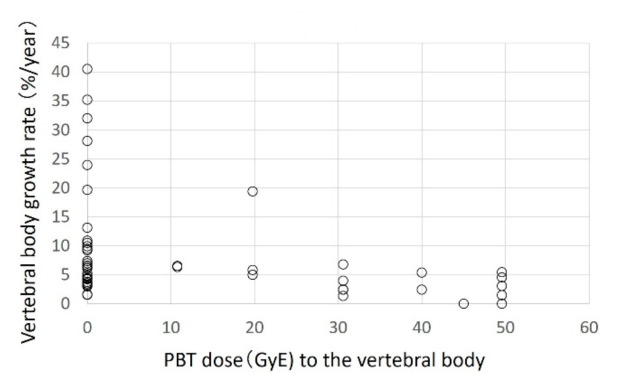
Scatter plot of the growth rate (%/year) of cervical vertebral bodies and proton beam therapy dose.

**Figure 2 cancers-13-00349-f002:**
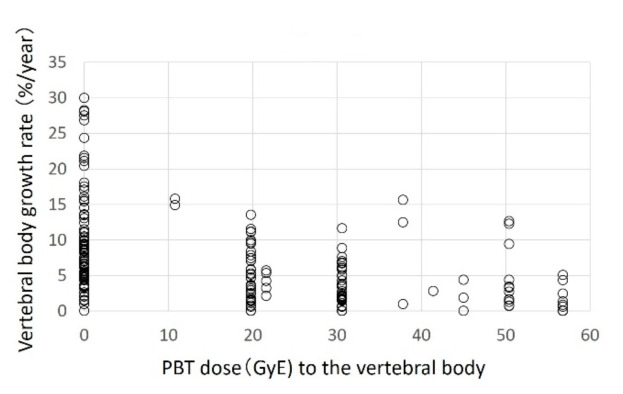
Thoracic vertebral bodies.

**Figure 3 cancers-13-00349-f003:**
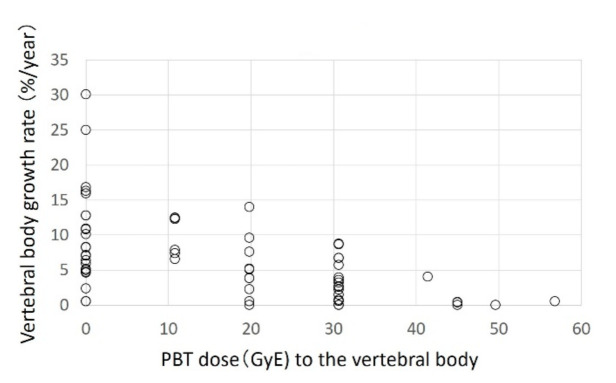
Lumbar vertebral bodies.

**Figure 4 cancers-13-00349-f004:**
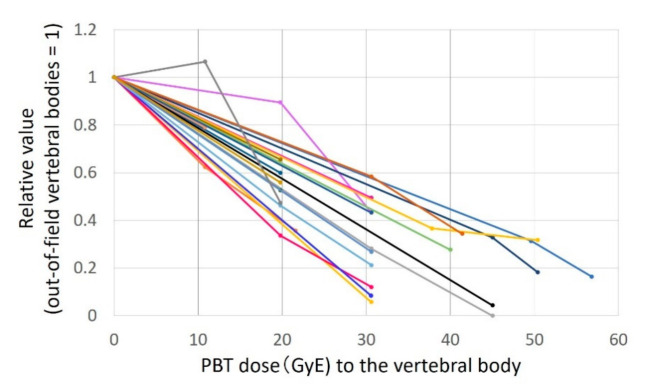
Relative growth rate of vertebral bodies compared with the control for each patient. Each line represents each patient and the vertical axis shows the ratio of the growth rate to out-of-field vertebral bodies.

**Figure 5 cancers-13-00349-f005:**
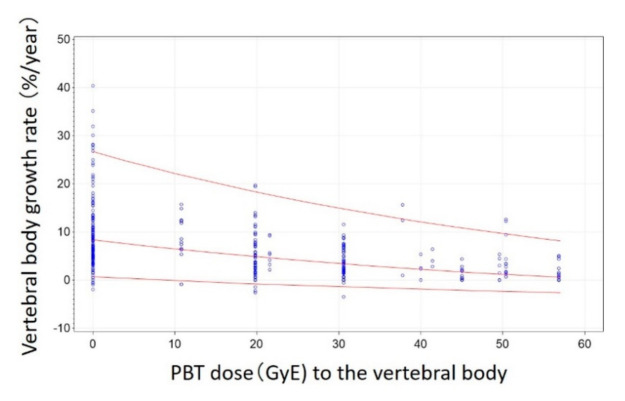
Scatter plot of the growth rate (%/year) of all measured vertebral bodies and proton beam therapy dose. Red lines represent 2.5, 50, and 97.5 percentile curves.

**Table 1 cancers-13-00349-t001:** Result of the fixed effects estimation with a best-subset selection.

Fixed Effect		Estimate	Standard Error	*t*-Value	*p*-Value
Age		−0.04854	0.03281	−1.48	0.1525
Sex	Female	−0.1799	0.1290	−1.39	0.1781
	Male	0			
Site of vertebrae	Cervical	0.1715	0.06332	2.71	0.0071
	Upper thoracic	0.06079	0.05449	1.12	0.2654
	Lower thoracic	0.05647	0.04893	1.15	0.2493
	Lumbar	0			
Dose		−0.01507	0.001246	−12.09	0.0006
Square of dose		0.000043	0.000099	0.44	0.6617
Interaction between the site and dose	Cervical	0.000645	0.003581	0.18	0.8572
	Upper thoracic	−0.00048	0.003292	−0.15	0.8833
	Lower thoracic	−0.00138	0.003105	−0.45	0.6565
	Lumbar	0			

**Table 2 cancers-13-00349-t002:** Characteristics of patients and tumors.

Item		Value
Median age (range)		4.4 (2.4–10.9)
Sex (male/female)		10/13
Disease		
Neuroblastoma		13
	Wilms’ tumor	3
	Ewing sarcoma	2
	Ependymoma	2
	Pilocytic astrocytoma	1
	Nasopharyngeal carcinoma	1
	Clear cell sarcoma of kidney	1
Median PBT Dose, Gy(RBE) (range)		30.6 (10.8–56.8)
Chemotherapy performed		22
History of the irradiation for head		2

PBT: Proton Beam Therapy.

## Data Availability

The data are not publicly available due to the facility’s privacy policy.

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
