# Peer review of "An Analysis of Vertebral Body Growth after Proton Beam Therapy for Pediatric Cancer"

_cancers, 2021, doi:10.3390/cancers13020349_

Round 1

Reviewer 1 Report

This article is a report of an analysis of vertebral body growth after proton beam therapy for pediatric cancer and shows that bone growth retardation occurred even at a low dose of 10 Gy, and that the growth rate linearly decreased as the dose increased without reaching a threshold value, so this article has interesting viewpoints. However, there are some comments and questions, so, minor revision is required.

  1. It is better to unify the representation. Childhood cancer or pediatric, second cancer or secondary cancer.
  2. Abstract L4; Proton beam therapy (proton beam therapy); It is not necessary in parentheses.
  3. L32; Space is not needed between Gy and (RBE).
  4. Introduction; P2L10; A period in required after et al.
  5. L70; RT is the first abbreviation, so it should be full spelled.
  6. L71 Proton; “P” should be lowercase. L88 Age; “A” should be lowercase.
  7. L126; What is CIS?
  8. L182, L196, Table 1; GyE should be change to Gy(RBE).

Author Response

Thank you very much for providing important comments. We are thankful for the time and energy you expended. 

According to your suggestion we revised our manuscript.

Your comment have helped us significantly improve our paper.

Reviewer 2 Report

This manuscript describes vertebral body growth after proton beam therapy for pediatric cancer patients. This is well written and well analyzed manuscript.

A Table for the results of multivariate analysis should be added for better understanding to journal’s readers.

Author Response

Thank you very much for providing important comments. We are thankful for the time and energy you expended. 

According to your suggestion we added a table for the results of multivariate analysis.

Your comment have helped us significantly improve our paper.

Regards,

Reviewer 3 Report

The paper «An analysis of vertebral body growth after proton beam therapy for pediatric cancer” is interesting and worthy of publication if major changes are made. The authors should be complemented for their methodology which in this reviewers’ opinion is good; comparison of vertebral growth intra- as well as interindividually is good. There are some major limitations, first and foremost the short observation period. This is, however, highlighted by the authors, and the findings showing growth impairment at an early stageas well as at low radiotherapy doses are interesting. Another limitation is the small number of patients and vertebrae investigated.

The English language needs extensive improvement; this pertains in particular to the introduction and the discussion. That is a prerequisite for publication. Examples are line 21 (“Old” should be deleted), and the sentence structure in lines 48-62 and 68-69.

The discussion should be shortened, especially the first paragraph.

Also, line 74: I would suggest to change from borderline significance to not significant (p-value of .120)

Line 126: Spelling error (CSI, not CIS)

The conclusion is sound, although I would suggest another wording than “conclude” as the limitations preclude the authors from drawing safe conclusions.

Author Response

Thank you very much for providing important comments. We are thankful for the time and energy you expended. Our responses to the referees’ comments are as follow:

The English language needs extensive improvement; this pertains in particular to the introduction and the discussion. That is a prerequisite for publication. Examples are line 21 (“Old” should be deleted), and the sentence structure in lines 48-62 and 68-69.

→We performed English proofreading by native speaker.

The discussion should be shortened, especially the first paragraph.

→We performed English proofreading and shortened discussion.

Also, line 74: I would suggest to change from borderline significance to not significant (p-value of .120)

Line 126: Spelling error (CSI, not CIS)

→We modified manuscript according to your suggestion.

The conclusion is sound, although I would suggest another wording than “conclude” as the limitations preclude the authors from drawing safe conclusions

→We change the ”conclude" more suitable expression.

Your comment have helped us significantly improve our paper.

Best Regards, 
